# Switched Reluctance Motor Design for a Mid-Drive E-Bike Application

**Mladen V. Terzić ***  **and Dragan S. Mihić**

School of Electrical Engineering, University of Belgrade, 11060 Belgrade, Serbia; dragan84m@etf.rs
* Correspondence: terzic@etf.bg.ac.rs; Tel.: +381-11-32-18-368

**Abstract:** The popularity of electric bicycles (e-bikes) among urban commuters and cyclists is constantly increasing because e-bikes provide an efficient, more powerful, and a low-cost mode of transportation. Since the main issue is to increase the power density of the drive system, a permanent magnet (PM) machine is the preferred choice. In mid-drive systems, higher speed motors are commonly used, which provides the opportunity to use switched reluctance machines (SRM), because they can provide better performance when designed for higher speeds. Moreover, the simple, robust, and low-cost structure of SRM makes it a favorable option for e-bike drive systems. In this paper, an SRM design for an e-bike mid-drive system is investigated. Several 3-phase configurations with higher number of rotor poles than the stator poles are considered: 6/10, 6/14, and 12/16, as well as the conventional 12/8 SRM, for the sake of comparison. Main dimensions and requirements are defined from Shimano Steps mid-drive PM machine, whose characteristics are taken as the design goal. According to the results, the best configuration is selected and further optimized, leading to the final design for which a prototype is built and tested.

**Keywords:** electric bicycles (e-bikes); switched reluctance motor; finite element analysis (FEA); machine design; mid-drive system



## 1. Introduction

The movement of the world toward sustainability increases the popularity of e-bikes, because they represent a light-weight, clean, and much cheaper mode of transportation. It was reported in [1] that a common commercial e-bike can travel a distance of about 20–30 km with single charge, which can save about CND $3.5 per charge or CND $1500 per year, when compared to a car and considering only the cost of gasoline [2]. The growing e-bike market, in China [3,4] and then in Western Europe [5] and USA, puts higher demand on the design of the whole e-bike drive systems, especially the electric motor as one of the major components. The main challenge is still to have a good tradeoff between the e-bike performance and the price. Thus, the main design requirements are to increase torque and power density, efficiency and robustness of the electric machine and to reduce its costs at the same time [6].

Sine- or square-wave excited permanent magnet motors are mostly used in e-bike drive systems [7–11]. With regard to the motor mounting position, e-bike drives are divided into two main groups: (1) in-wheel mounted hub motors and (2) frame mounted mid-drive motors. In the mid-drive systems, brushless DC (BLDC) motor is the only technology that has been used so far. Existing mid-drive systems with their main properties are given in Table 1 [12–23]. From the given data, nominal power is 250 W with electrical assistance up to 25 km/h in accordance with the European regulations [24]. Difference comes from the higher allowed power and speed in the countries like Canada and the United States [24]. For these countries, maximum allowed speed is 32 km/h and the power 500 W and 750 W, respectively. Despite their benefits in terms of torque assistance and higher speed, e-bikes still have high price as compared to conventional bicycles. The

cost of an electric bicycle with Shimano mid-drive system is around $2600. Other higher performance variants usually cost over $4000 [20]. The other problem related to e-bikes might be their heavy weight. According to the e-bike reviews, the weight of e-bikes with mid-drive systems is usually around 22 kg. To decrease the price and the weight at the same time, new motor drive solutions must be developed. In that sense, a switched reluctance machine is an attractive candidate for e-bike applications. Because mid-drives are always equipped with a gear-set, the motors usually operate at higher speeds than the motors used for in-wheel hub drives. Higher-speed operation of mid-drive motors also enables the utilization of a switched reluctance motor (SRM) for this application because it becomes more competitive with an interior permanent magnet motor (IPM) at elevated speeds. Furthermore, its simple and robust structure, as well as low production cost gives SRM an advantage over permanent magnet motors. On the other hand, an SRM converter tends to be more expensive due to higher number of semiconductor devices and larger DC link capacitance. However, it is hard to give a general comparison because it highly depends on the application for which the drive is designed for. Furthermore, the converter and the SRM must be designed together in order to achieve optimal drive for the set of design goals and restrictions. In addition to other benefits for mid-drive application, the utilization of the SRM configuration with higher number of rotor poles than the stator poles offers higher power density and lower torque ripples than in configurations with lower rotor than stator poles [25–27]. So far, no SRM designs for mid-drive e-bike system have been reported. Only solutions for in-wheel mounted hub motors were investigated, like in [28]. This solution is with external rotor SRM and for much lower speeds than can be expected in the mid-drive system. Thus, design of an SRM for the mid-drive is essentially different and much more challenging because of the strict space requirements which impose much higher target for the torque density.

**Table 1.** Main characteristics of existing mid-drive e-bike systems.

| Drive Type | Nominal Power [W] | Max Power [W] | Nominal Torque [Nm] | Max Torque [Nm] | Maximum Support Speed [km/h] | Weight [kg] |
|---|---|---|---|---|---|---|
| BOSCH Performance line CX [14–16] | 250 | 500–950 | 20 | 40–75 | 25–45 | 3.8 |
| YAMAHA (EXPW mode) [15–17] | 250 | 1000 | 20 | 80 | 25 | 3.1 |
| Shimano Steps [15,16,18] | 250 | 500 | 25 | 50 | 25 | 3.19 |
| BROSE E45 [15,16,19] | 250 | 1130 | 28 | 90 | 25–45 | 3.4 |
| IMPULSE 3.0 [16,20–22] | 250 | 1257 | 20 | 100 | 25 | 3.81 |
| TranzX M16 [16,23] | 250/350 | - | - | 50–65 | - | 3.9 |
| 8Fun BBS01 [24,25] | 350 | 680 | 40 | 80 | 32 | 3.62 |
| 8Fun BBS02 [16,24,25] | 750 | 1300 | 80 | 128 | 32 | 4.08 |

The aim of this paper is to introduce the design of SRM for a mid-drive e-bike application. Several topologies are investigated and designed. Namely, one conventional 12/8 configuration and three other configurations with higher number of rotor poles than stator poles have been investigated: 6/10, 6/14, and 12/16. Section 2 starts with definition of the design requirements, in which purpose Shimano Steps IPM is inspected in detail using FEA software and its characteristics are presented in detail and are then taken as a design goal for all SRM configurations. Following this, the design of the chosen four SRM topologies has been conducted and the main results are presented and compared. Based on the comparison of the main results, the best configuration is chosen and analyzed in more detail. Section 3 presents the prototype of the final design, which is built and tested, and comparison of the results to the calculations. Finally, the Section 4 summarizes all the results and provides the major conclusions.

## 2. Design Methodology

### 2.1. Shimano Steps Motor Benchmarking

To define the input design parameters and requirements, a Shimano Steps mid-drive unit has been disassembled and the electric motor has been analyzed in detail. The motor is analyzed through series of transient simulations using Ansys Maxwell FEA software.

#### 2.1.1. The Geometry and Operational Parameters of Benchmark IPM Motor

After we disassemble the mid-drive system, the motor is taken and inspected in detail. The main dimensions and some important parameters of the winding are summarized in Table 2.

**Table 2.** Main parameters of the Shimano Steps drive unit.

| Parameter | Value |
|---|---|
| Nominal/maximum power [W] | 250/500 |
| Nominal/maximum torque of the drive unit [Nm] | 25/50 |
| Battery voltage [V] | 36 |
| Gear ratio | 37.3 |
| Outer diameter without housing [mm] | 76 |
| Shaft diameter [mm] | 12 |
| Lamination stack length [mm] | 20.9 |
| Motor housing length [mm] | 40 |
| Number of stator slots | 12 |
| Number of rotor poles | 14 |
| Number of turns per coil | 23 |
| Wire gauge [AWG] | 18 |
| Phase resistance at 20 °C [Ω] | 0.1349 |
| Magnet type | NdFeB |
| Steel Sheet Thickness [mm] | 0.35 |

Motor from disassembled drive is shown in Figure 1. The mid-drive motor is an interior permanent magnet machine, so called flux focusing or spoke type design. Motor has 12 slots and 14 poles, and the winding is realized as fractional slot, tooth pitched. Phase windings are connected in delta, and their layout is shown in Figure 2.

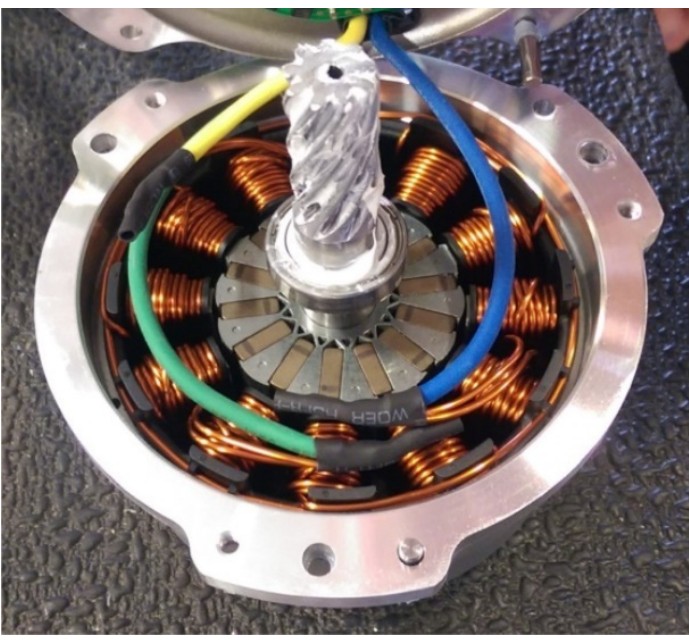

**Figure 1.** IPM from Shimano Steps mid-drive bike system.

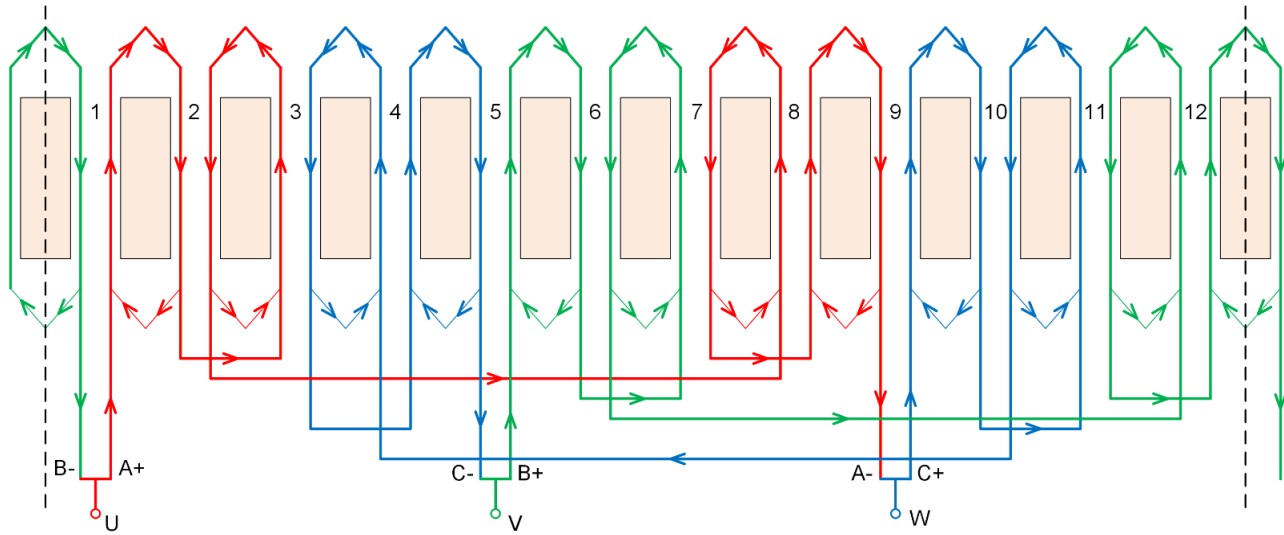

**Figure 2.** Shimano Steps motor winding layout.

The motor is supplied from a three-phase inverter, and it is driven like sine wave drive. The DC link voltage is equal to the battery voltage, which is 36 V in the Shimano Steps mid-drive system. Therefore, the maximum value of the first harmonic phase voltage is:

$$V_{\max} = \frac{\sqrt{3}}{2} V_{DC} = 31.18V, \tag{1}$$

This voltage was taken as a boundary value between constant torque and flux weakening regimes during the characterization.

### 2.1.2. Performance Analysis

According to the given parameters and measured geometry, 2D FEA model is created and simulated using Ansys Maxwell FEA software. Typical material properties have been used for the characterization. For the laminations 35JNE300 material is chosen, which is typically used for small precision and traction applications. The knee point of the magnetization curve of 35JNE300 is around 1.8 T. For the magnets, NOMAX-42 (Sintered NdFeB irreversible) is chosen, which has the coercivity of 1,004,195 A/m, relative permeability of $\mu_r = 1.0492$, and remanence of $B_r = 1.324$ T.

First, nominal (base) speed is obtained through several transient simulations, which are set according to the given maximum power and $V_{\max}$ derived from (1). The aim of transient FEM simulations was to precisely obtain the base motor speed at which it must enter the flux weakening regime. This is the speed at which $V_{\max}$ is reached and the only way to further increase the speed is to apply negative d-component of the current vector and thus to perform flux weakening. This speed can be only determined from the transient simulations by checking if the sum of EMF and internal voltage drops (on the motor resistance and inductances) exceeds $V_{\max}$ or not.

For each simulation, constant rotational speed is set, and motor is supplied with three-phase current with controllable amplitude and phase angle. After each simulation, fast Fourier transformation (FFT) is performed on the phase voltage waveform and the maximum value of the first harmonic is compared to $V_{\max}$. All simulations are performed for one full mechanical cycle to have enough points for FFT. Average torque is also evaluated and used for the calculation of output power which is then compared to the given maximum power. In this way, nominal speed is found to be approximately 3000 rpm and maximum RMS line current of about 14 A. According to the nominal power and obtained nominal speed, nominal RMS line current is found to be around 7 A. For each current, excitation current angle ($\gamma$ angle) has to be adopted in order to maximize the output torque. This

angle represents the phase difference between the induced EMF vector and the excitation current vector, and it is directly related to the electromagnetic torque $T_{em}$:

$$T_{em} = \frac{3}{2}p\left(\Psi_{PM}I\cos\gamma + \frac{1}{2}(L_d - L_q)I^2\sin 2\gamma\right), L_d > L_q, \tag{2}$$

where $p$ is number of pole pairs, $\Psi_{PM}$ is the magnet flux, $L_d$ and $L_q$ are d- and q-axis inductances, and $I$ is the RMS phase current. Figure 3 shows the change of torque with respect to the excitation current angle $\gamma$ at the nominal current. It can be observed that, at the nominal current, the maximum torque of 0.802 Nm is achieved with 6.45°. Small value of optimum $\gamma$ angle is due to the very small reluctance in this machine which comes from the small difference between $L_d$ and $L_q$ inductances. To prove this statement, these inductances are obtained from FEA simulations at nominal current and are given in Table 3.

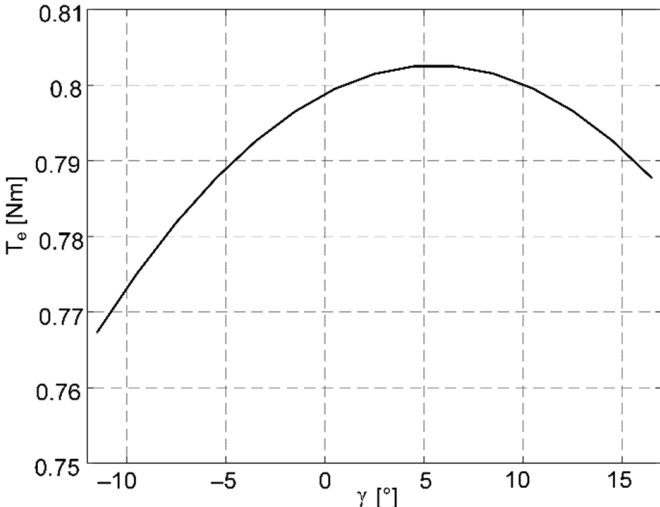

**Figure 3.** Change of torque with different excitation angles at nominal current.

**Table 3.** Motor parameters at nominal speed, at nominal ($I_n$) and maximum current ($I_{max}$).

| Parameter | $I_n$ | $I_{max}$ |
|---|---|---|
| Excitation (line) current RMS [A] | 7.07 | 14.14 |
| Phase current RMS [A] | 4.08 | 8.16 |
| Current density [A/mm²] | 4.96 | 9.92 |
| Excitation current angle, $\gamma$ [°] | 6.45 | 8.64 |
| Induced line-to-line voltage (1st harmonic) [V] | 30.29 | 31.14 |
| Speed [rpm] | 3047 | 3047 |
| Torque [Nm] | 0.802 | 1.592 |
| Output Power [W] | 256 | 508 |
| Copper loss [W] | 6.96 | 27.3 |
| Core loss [W] | 10.36 | 10.48 |
| Efficiency [%] | 93.66 | 93.07 |
| $L_d/L_q$ inductances at nominal current [mH] | 109.2/101.2 | - |
| Torque density [Nm/kg] | 1.521 | - |

Knowing all these parameters, simulations have been performed for the nominal and maximum currents with appropriate conduction angles at nominal speed. As a result, electromagnetic torque, copper, and iron losses are found and used for the calculation of input and output power, and efficiency. In the calculations, losses in the magnets are neglected. Table 3 summarizes all important simulation results. Last row in Table 3 contains torque density value which will be compared with the values of SRM designs later on.

Figure 4 shows torque versus time at the nominal speed and current. Using maximum, $T_{\max}$, and average torque, $T_{avg}$, torque ripple value, $T_{ripple}$, is calculated as:

$$T_{ripple} = \frac{T_{\max} - T_{avg}}{T_{avg}}\ [\%],\qquad(3)$$

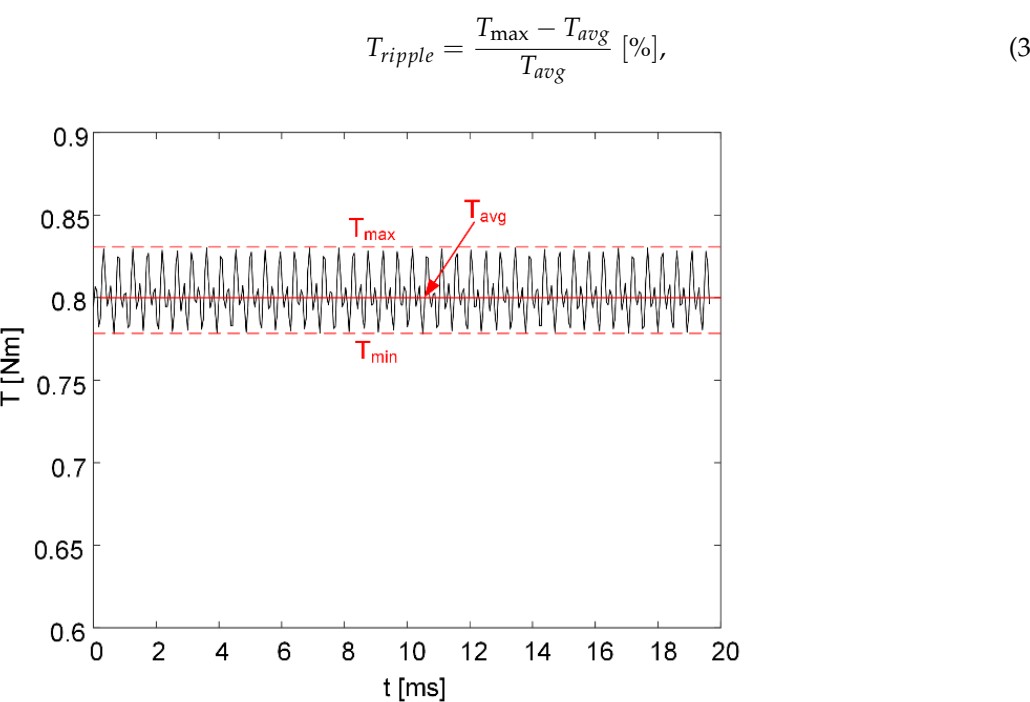

**Figure 4.** Motor torque at nominal current and optimum gamma angle.

Flux density in the air gap at different current levels is shown on Figure 5. It can be observed that current level does not affect flux density level in the motor too much, which can also be seen from Figure 6. This makes the core losses of the motor almost independent of the excitation current. From these results, it can also be concluded that the armature reaction in this motor is very small. This is due to the small d-axis inductance which requires very high currents to produce considerable armature reaction. This also reduces flux weakening capabilities of this motor and makes the flux weakening region very narrow.

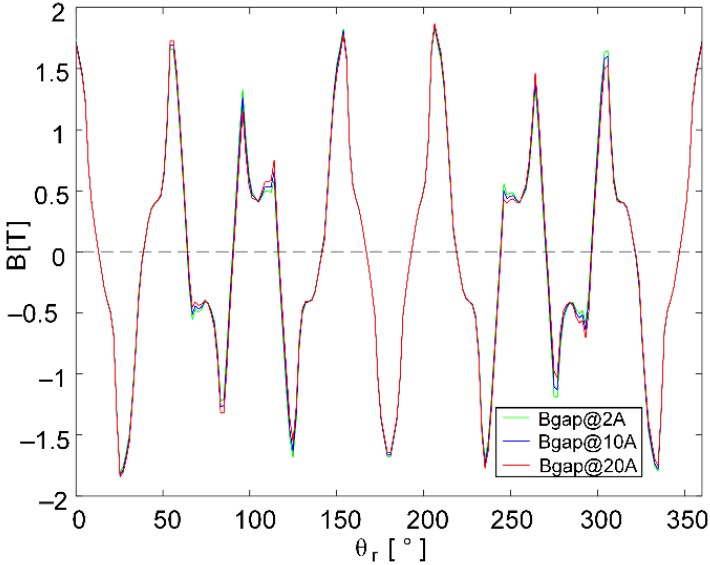

**Figure 5.** Flux density in the air gap at different supply currents.

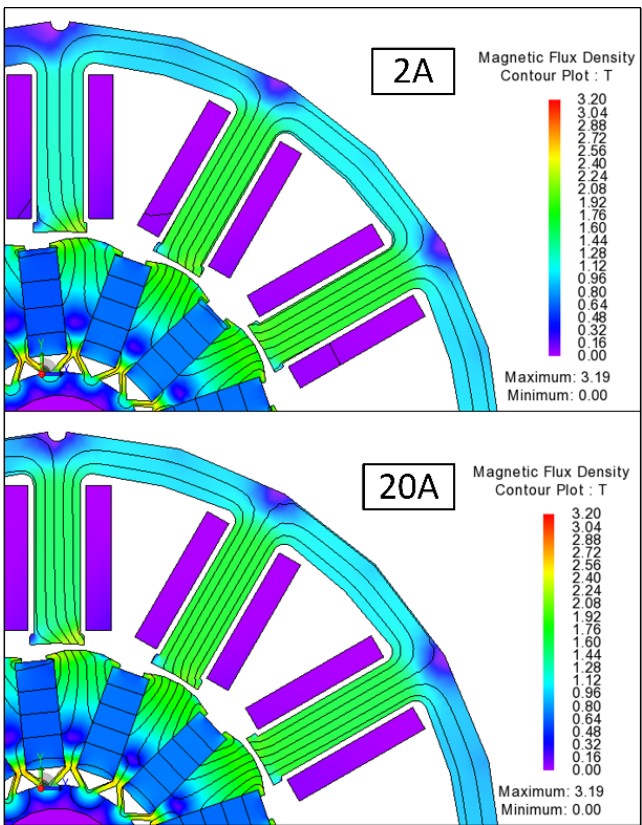

**Figure 6.** Flux density plots from FEA software with different excitation currents and at the same rotor position.

Having defined the nominal and maximum values, we can now perform simulations for other operation points. Simulations are performed at different speeds and different current levels in order to obtain torque-speed curve, loss and efficiency maps. Figures 7–10 present the most important results of this study, which will be used later as a design goal in SRM design process.

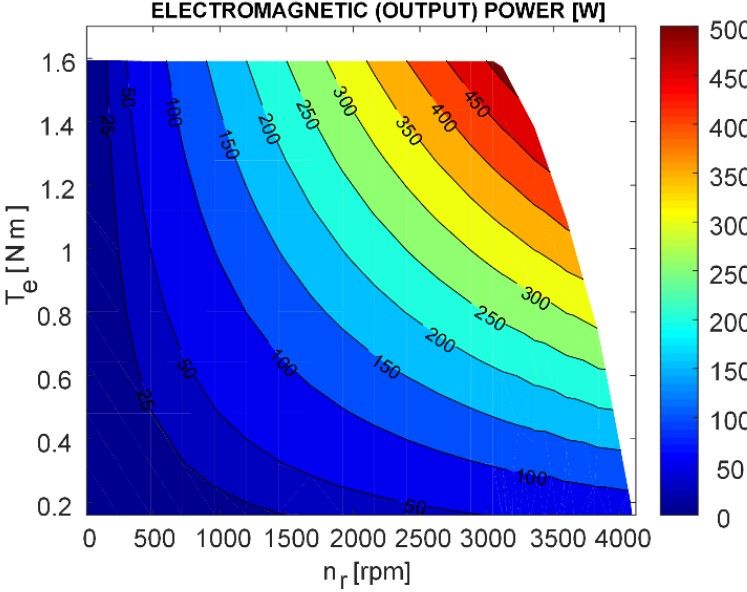

**Figure 7.** Shimano Steps IPM output power.

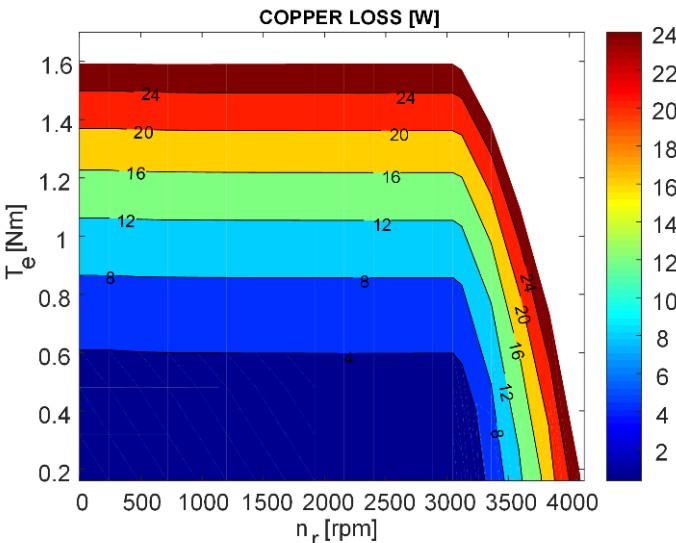

**Figure 8.** Shimano Steps IPM stator copper losses.

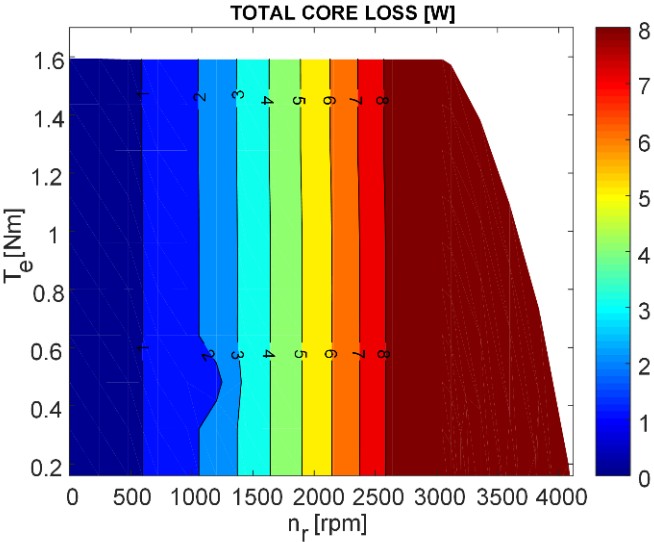

**Figure 9.** Shimano Steps IPM stator and rotor iron losses.

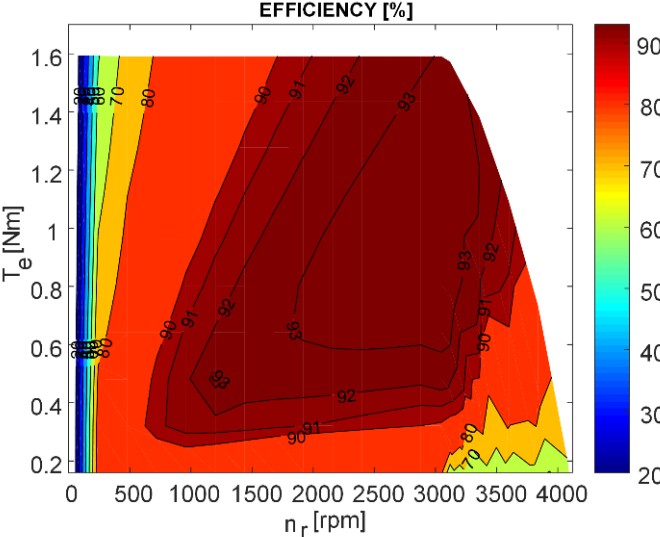

**Figure 10.** Shimano Steps IPM efficiency map.

As it can be seen from Figure 7, maximum power that can be extracted from this motor is 500 W. The flux-weakening region is very narrow which leads to the conclusion that this mid-drive e-bike motor is designed to operate mainly in constant torque region. The maximum speed at which the motor can still deliver nominal power is around 3900 rpm which corresponds to 100 rpm cadence of pedals.

Copper and total core losses are given in Figures 8 and 9. Core loss does not change with the current level. The main reason for this is the fact that the armature reaction is very small, and the flux density is almost the same at different currents, as previously stated and concluded from Figure 6.

As it can be seen from the efficiency map in Figure 10, the mid-drive e-bike motor has a wide high efficiency region above 90%. It must be mentioned that the efficiency is calculated without taking the mechanical and magnet losses into account. In the largest part of the operating region, torque ripple is below 4%, which is a good property of IPM in general.

According to the given properties and parameters of IPM, design requirements and constraints for the SRM design will be defined in the next subsection. Then, the design procedure will be described and demonstrated on several different SRM configurations.

*2.2. SRM Design*

In this subsection, main steps in SRM design for the mid-drive electrical bicycle system are described. Design is based on the previously found properties of Shimano Steps IPM.

### 2.2.1. Input Parameters and Design Constraints

Using the main dimensions of the IPM, in Table 4 we define and present the main geometrical constraints for the SRM design. In order to get one-on-one comparison between designs, main dimensions, like outer diameter and stator stack length, are kept the same as in IPM. Maximum total length of the SRM, which means stator stack length plus the length of the end-windings, is limited to the length of the original housing in which IPM is placed. These geometry constraints ensure that SRM can fit in the same case as IPM.

**Table 4.** Main geometrical constraints.

| Name | Value |
|---|---|
| Maximum outer diameter [mm] | 76 |
| Maximum stack length [mm] | 20.9 |
| Maximum total length [mm] | 40 |
| Minimum shaft diameter [mm] | 12 |
| AWG/Cross section [mm$^2$] | 18/0.823 |

Main electrical design constraints are summarized in Table 5 and their choice is described briefly in the following. For this application it is necessary to keep the same rating of the power supply, so the DC bus voltage and converter output line RMS current are not changed. Thus, the phase RMS current of the SRM can be higher by $\sqrt{3}$ than in IPM because windings of the IPM are in delta connection while in SRM the phases are supplied separately. Because the wire size is the same, current density in SRM will be higher than in the IPM, but still not much higher than maximum permissible value for the AWG 18 at the nominal current, which is 7.75 A/mm$^2$. At the end of the design procedure, thermal analysis will be performed for each configuration in order to check the maximum temperature rise of the windings and apply some corrections to the initial values of the currents if necessary.

For the SRM design, it is also important to know the commanded peak current for the chopping control. This value is usually 50–70% higher than the phase RMS current, depending on the operation regime. By taking the factor of 1.7, the current chopping limit is calculated and given in Table 5, both for the maximum and the nominal operating regimes.

**Table 5.** Main electrical constraints.

| Parameter | Regime | |
| --- | --- | --- |
| | Nominal | Maximum |
| Phase current (RMS value) [A] | 7.07 | 14.14 |
| Commanded peak current [A] | 12 | 24 |
| Predicted current density [A/mm$^2$] | 8.59 | 17.2 |
| Battery voltage [V] | 36 | 36 |
| Base speed [rpm] | 3047 | 3047 |
| Desired torque at base speed [Nm] | 0.8 | 1.59 |
| Desired output power [W] | 250 | 500 |

In addition to the parameters in Table 5, targeted torque-speed curves for both nominal and maximal current operation are given in Figure 11. These curves will be used also for comparison of SRM torque-speed curves.

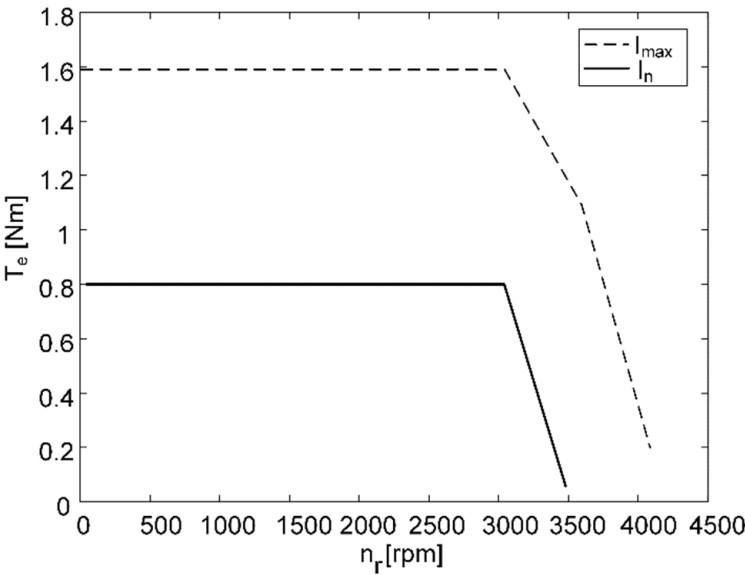

**Figure 11.** Target torque-speed curves at nominal ($I_n$) and maximal ($I_{max}$) current.

2.2.2. Design and Optimization Procedure

After defining the main constraints and requirements, we can initiate the SRM design procedure. Since main motor dimensions are defined from the requirements, it was not necessary to perform initial machine sizing like in usual design procedures. As the SRM is highly nonlinear in its operation there is no unique analytical calculation procedure which can give trusted results for all SRM configurations the hybrid calculation method (FEM plus analytic) is applied, and it will be explained in the following.

Thus, a very first step in the SRM design is the choice of the suitable number of stator and rotor poles. According to the IPM benchmark design which has 12 slots, design of SRM should start with the 12-stator pole configuration. Since the number of stator poles (teeth) is the same, the adjustment of the 12-stator pole SRM geometry should require the shortest iteration process. Between the two 12 stator pole configurations, the conventional one with the 8 rotor poles is chosen for the start. Designs with the higher number of rotor poles can ensure lower torque ripple and possibly higher torque density as stated in [25–27]. For this reason, 12/16 SRM configuration is taken as the second 12-stator-pole configuration. Although higher rotor pole number increases the torque ripple frequency proportionally, benefiting the average output torque, this might come at the expense of higher losses and, hence, lower efficiency [29]. Due to this reason, and the restricted space of the motor, number of rotor poles higher than 16 might not be suitable for this application.

One of the main tasks when designing SRM to compete with the IPM is to provide enough magneto-motive force (MMF) from the coils to compensate the lack of MMF from the magnets. 12-stator-pole configurations might not have enough space in the slot for achieving this goal. Thus, configurations with the lower number of stator poles, such as 6, will be considered as well. In [26,27,30], it was shown that 6/10 configuration has higher torque density and lower torque ripple when compared with a conventional 6/4 or 6/8 SRM. That is why 6/10 SRM will also be investigated for the mid-drive application. To explore the possible advantage of higher number of rotor poles, a 6/14 SRM will also be designed.

After number of poles is decided, restrictions regarding permissible pole arc lengths must be calculated for each configuration, according to the following equations [28]:

$$
\begin{cases}
\min(\beta_s, \beta_r) \geq \frac{360}{mP_r} \\
\beta_s + \beta_r < \frac{360}{P_r}
\end{cases} ,
\tag{4}
$$

where $m$ is the number of phases, $\beta_s$ and $\beta_r$ are pole arc angles of stator and rotor, respectively, and $P_s$ and $P_r$ are the number of stator and rotor poles, respectively. It is important to fulfill these requirements to ensure self-starting capability of SRM.

The design procedure is basically iterative and starts once the main dimensions and requirements are defined. Design flow chart of the e-Bike SRM is shown in Figure 12. The SRM design was carried out using a design-space exploration method, where each geometry parameter combination was swept. Stator and rotor pole and yoke dimensions are adjusted for each configuration, along with the number of turns per pole and the slot fill factor. During these changes stator outer diameter and stack length were kept constant due to the design requirements.

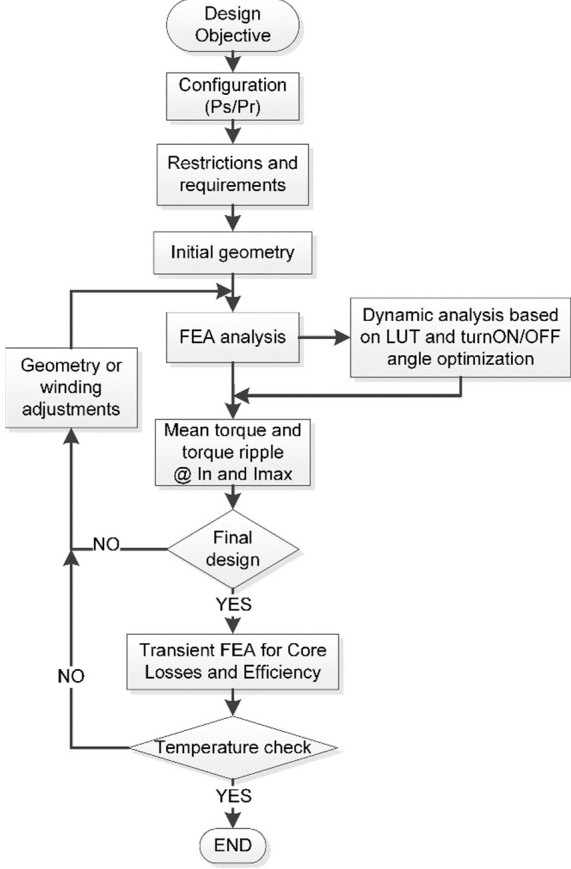

**Figure 12.** SRM design flow chart (LUT-look up table; $I_n$ and $I_{max}$-nominal and maximum current values, respectively).

The pole arc angles were swept in increments of 0.1°, while stator and rotor poles height were swept in increments of 0.5 mm. The wire size was chosen from the catalogue of standard enameled copper wires and was adjusted while maintaining a target number of turns and a maximum slot-wire fill factor of 0.58. The number of coil turns was varied for all designs to find the correct balance between copper loss, EMF, and power factor. It is desirable to use as many turns as necessary to achieve the desired torque requirements at low speeds while maintaining a low enough voltage to enable enough torque in flux weakening region. This will also give voltage drop and copper loss as low as possible. Due to mechanical reasons the airgap was kept higher than 0.4 mm.

When a given parameter is swept, all other parameters are kept constant, allowing evaluation of each possible combination of parameters in the swept region. The parameter sweep ranges were bounded depending on defined geometrical design constraints (Table 4). The ranges were also limited to realistic values and the limits were confirmed through the parameter sweep analysis itself.

The first step in the design process uses a series of constant current 2D FEA analyses in software Ansys Maxwell. In the constant-current analysis, one phase is excited with maximum phase current at a fixed rotation speed to determine the phase torque and voltage at that speed. This gives the phase torque profile of the motor and a rough indication of the base speed. The number of designs is then reduced to the set of highest performing designs.

In order to find electromagnetic torque and other important parameters for certain geometry, FEA software Ansys Maxwell is applied for dynamic simulations with constant current excitation. Results of these simulations are the torque/flux profiles in the form of Look Up Tables (LUTs), which are than modeled in MATLAB/Simulink, with classic turn-ON and turn-OFF angular position control in order to get machine dynamic performance [31,32]. The dynamic-current analysis gives the total torque and phase voltage when phase overlap is accounted for. Optimum turn-ON/OFF angles are found by multi-objective genetic algorithm for each operational point. In this way, the maximum output torque with minimum torque ripple is obtained. Comparison between different variants is done on the basis of mean torque values when motor operates at nominal and maximum current and at nominal speed. These results are used to search for the final design inside the optimization algorithm which has ratio of mean torque and torque ripple as the cost function. In a single optimization iteration new population is made by changing the motor dimensions and winding parameters (number of turns and wire size). Finally, a geometry sensitivity analysis is conducted on the best dynamically performing design to fine-tune the parameters.

Since the model which is based on LUTs does not include loss calculation, after the final geometry was obtained, transient (time-step) FEA simulations in Ansys Maxwell software are performed to find the losses and efficiency and to check the torque values which were obtained from dynamic simulations using LUTs.

Finally, for all motor variants thermal analysis are performed for several operating points in order to check the maximum temperature in the motor. Thermal simulations are conducted using Motor-CAD software. Insulation system is considered to be in H class with maximum permissible temperature of 180 °C. Operation of e-bike motor is intermittent. This means that machine operates in one regime for a limited amount of time, so it usually does not have enough time to warm up to the steady state temperature. If one common e-bike driving cycle [28] is inspected, it can be concluded that for the nominal operation regime one hour can be taken as a maximum continuous operation time. Thus, temperature at the end of one-hour period for the nominal operational regime is taken for the maximum temperature. For the operation at the maximum current, temperature at the end of 100-s period can be taken as a limit. This is actually very long period for maximum operational regime, but it is taken for safety reasons.

In the following, the main results for the final solutions of different configurations are compared in terms of mean torque, torque ripple, losses, efficiency, and maximum temperature.

### 2.2.3. Comparison of Final Solutions for Different Configurations

After optimal configurations have been found, transient electromagnetic FEA and thermal simulations were performed for several operational points, at different currents and speeds, in order to get all necessary data for comparison. Table 6 gives main geometry properties of final designs for different configurations.

**Table 6.** Main geometrical properties of final SRM designs.

| Name | 12/8 | 12/16 | 6/10 | 6/14 |
|------|------|-------|------|------|
| Stator outer diameter [mm] | 76 | 76 | 76 | 76 |
| Rotor outer diameter [mm] | 38.6 | 48.4 | 54.6 | 57.2 |
| Lamination stack length [mm] | 20.92 | 20.92 | 20.92 | 20.92 |
| End turn length (both sides) [mm] | 8.74 | 10.58 | 19.77 | 21.61 |
| Total axial length [mm] | 29.66 | 31.5 | 40.69 | 42.53 |
| Number of turns per pole | 43 | 38 | 58 | 51 |
| Stator pole arc angle [°] | 15 | 7.9 | 13 | 9 |
| Rotor pole arc angle [°] | 15 | 7.9 | 13 | 9 |
| Wire fill factor | 0.58 | 0.57 | 0.58 | 0.57 |
| Stator pole/yoke ratio | 4.23 | 2.77 | 1.94 | 1.57 |
| Rotor pole/yoke ratio | 1.29 | 1.29 | 1.20 | 1.20 |
| Torque density [Nm/kg] | 0.971 | 1.097 | 1.223 | 1.091 |

As mentioned before, first 12/8 SRM was designed and optimized to achieve the highest possible torque. After that, design of 12/16 SRM was initiated by increasing just the number of rotor poles to 16. This naturally reduced the aligned to unaligned inductance ratio and thus the maximum torque per one stroke as well as mean torque value per one electrical cycle, as stated in [25]. To recover the torque value, rotor radius has to be increased to compensate in one way for the lost torque. This in turn reduces the available space for the winding which can be compensated to a certain extent by reducing the stator pole arc length. Stator outer diameter and yoke thickness could not be changed because of motor space requirements and mechanical rigidity, respectively. Following this procedure, optimal 12/16 SRM configuration was obtained with the parameters that are given in Table 6.

In 6/10 SRM, because the winding space is now doubled, pole height can be considerable reduced. After achieving geometry with the highest possible torque, final tuning of pole arc lengths has been done to reduce the torque ripple as much as possible. Finally, 6/14 SRM was designed following up the same principle as for the transition from 12/8 to 12/16 configuration.

Figure 13 shows the average values of torque for different SRM configurations in comparison to IPM for several speeds and at nominal current. Nominal chopping current limit for all SRM is 12 A, while the nominal RMS line current for IPM is 10 A. It can be seen that 6/10 configuration has higher torque at all speeds in the given range and it almost catches the IPM at low speeds. This is mainly due to the fact that 6/10 SRM has the highest number of turns per pole, as can be seen from Table 6, and thus the highest MMF. Configurations 12/8 and 6/14 express nearly the same behavior, while 12/16 has little bit lower torque when compared to them. Each SRM configuration has much better performance than Shimano Steps IPM in the region above the nominal speed. Last row in Table 6 shows torque densities for all SRM designs and as can be seen 6/10 has the highest value of 1.223 but still lower than IPM torque density of 1.521 which is given in Table 3.

Figure 14 depicts change in the aligned to unaligned inductance ratio with current. It gives a better understanding of the differences between the torque-speed curves of the SRM configurations. This ratio determines the capability of SRM to produce torque [25]. As can be seen from Figure 14, 6/10 aligned to unaligned inductance ratio is the highest for all currents in the range. This explains why the 6/10 SRM has the highest produced torque in the entire speed range. As it was stated before, average torque depends also on the MMF, which is the highest in the 6/10 SRM. Number of strokes per revolution also contributes to torque and this should give advantage to the solutions with the higher number of poles.

But this gives benefits only if it does not deteriorate the inductance ratio too much. This fact explains also why 6/14 has lower performance than 6/10 configuration even if 6/14 has the highest outer rotor diameter.

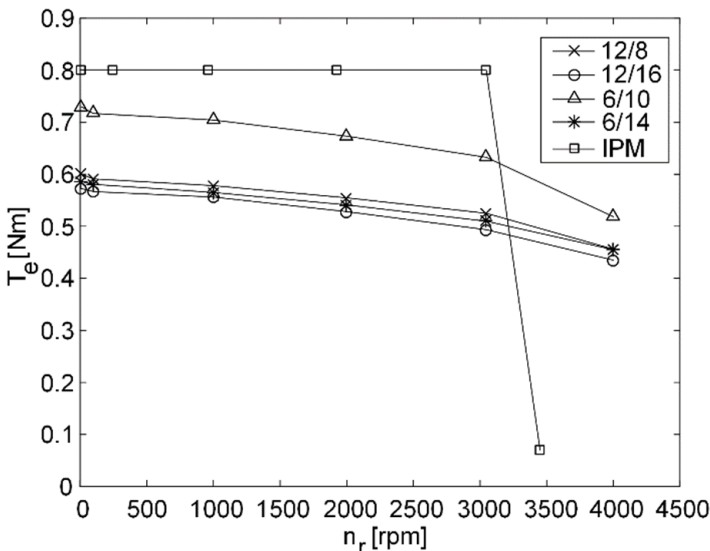

**Figure 13.** Torque-speed curves for all SRM configurations and the IPM at the nominal currents.

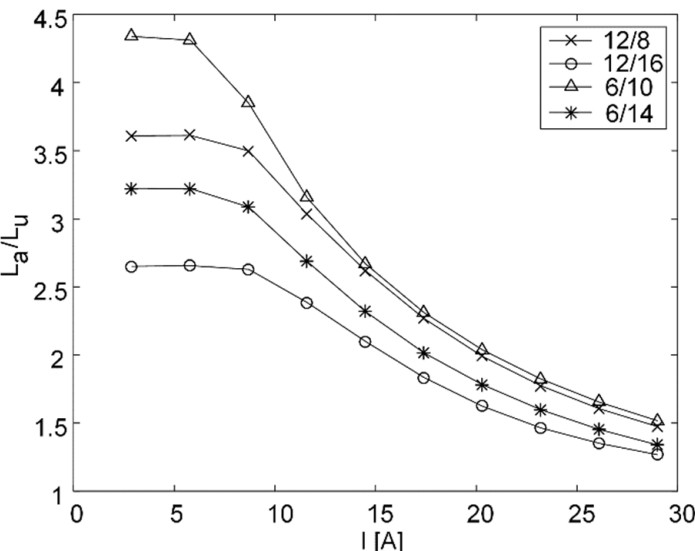

**Figure 14.** Aligned to unaligned inductance ratio for different SRM configurations and at different currents.

In SRM design, it is also of great importance to have the lowest as possible ratio between pole height and yoke thickness to increase mechanical rigidity and attenuate noise and vibration (NV) problems. Regarding this it can be seen from Table 6 that 6/14 SRM has the lowest ratio of pole height and yoke thickness at both rotor and stator side. This makes it the most rigid and the most preferable solution regarding NV.

The comparison between torque-speed curves at maximum current is given in Figure 15. The line current for IPM is 20 A and the windings are delta connected. As explained previously, to match with the phase current of SRM with the line current of IPM, maximum current chopping limit is set to 24 A. Like the nominal current case, 6/10 configuration has the closest torque-speed curve to that of IPM.

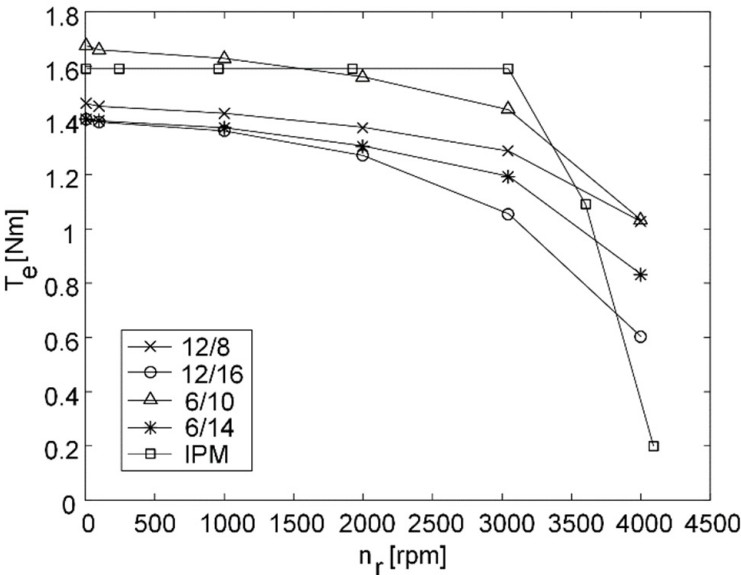

**Figure 15.** Torque-speed curves for all SRM configurations and the IPM at the maximum currents.

The comparisons between RMS values of torque ripples for different configurations at nominal and maximum current operation are given on Figures 16 and 17, respectively. All SRM configurations have nearly the same torque ripple values, while operating at nominal current, except at high speed where 6/10 SRM has the largest value. It is interesting to note that 6/14 SRM has much lower torque ripple in the maximum current operational regime than other designs, except at very high speeds. High torque ripple is the inherent property of SRM so in this field it can hardly be comparable to IPM. However, during the operation at nominal current and below nominal speed SRM behaves well, having RMS torque ripple values between 0.02 and 0.05 Nm at mean torque values between 0.6 and 0.7 Nm. In addition to this, at lower speeds torque sharing functions can be applied to reduce torque ripple values even more [33,34]. Moreover, torque ripple is not the root cause of NV problems in electrical machines thus obtained values does not necessarily mean that SRM motor will be much noisier than the IPM. However, detailed analysis of NV behavior is out of the scope of this work and is left for future studies.

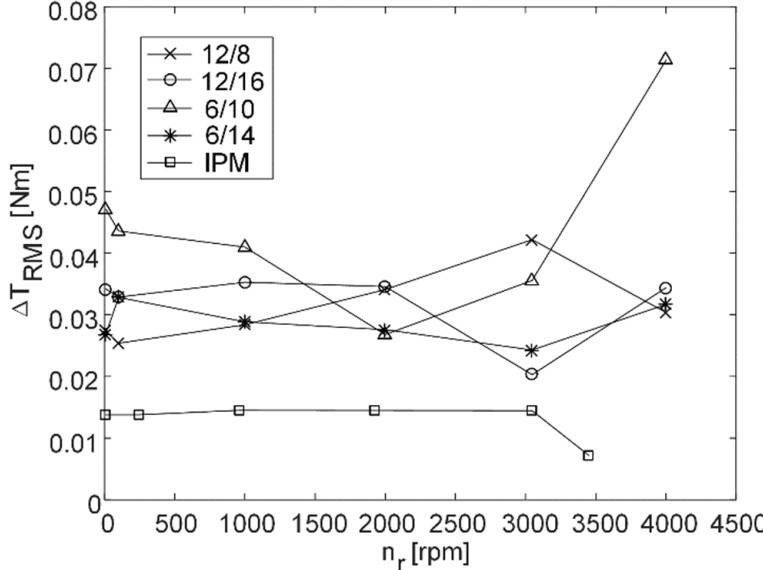

**Figure 16.** Torque ripple values at different speeds and nominal current.

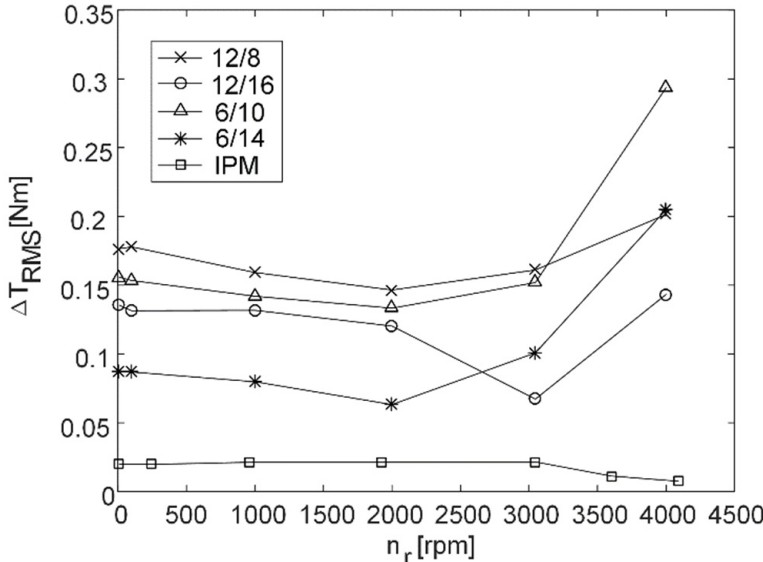

**Figure 17.** Torque ripple values at different speeds and maximum current.

Figures 18 and 19 present results of copper and core losses, respectively, when motor operates with the nominal current. As it can be seen from Figure 18, copper losses are much higher for each SRM design when compared to IPM, because of the higher stator MMF in SRM. Among different SRM configurations, 6/14 has the lowest copper loss but on contrast the highest core losses. Core losses in different SRM designs are more comparable with IPM than copper losses. It is interesting to note that 12/8 configuration has the lowest core losses because of the lowest switching frequency.

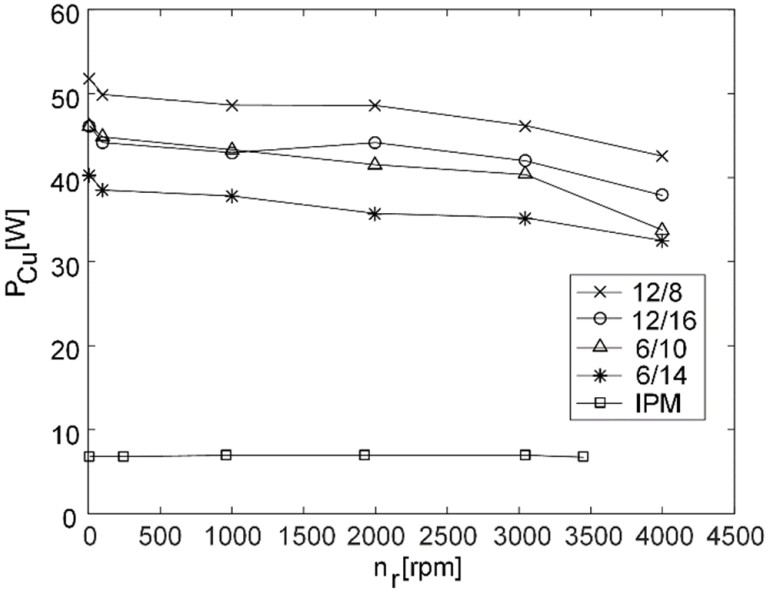

**Figure 18.** Copper losses at nominal current operation.

Efficiencies of IPM and all SRM designs are compared in Figures 20 and 21 for the nominal and maximum current, respectively. Because of the higher copper losses in SRM, efficiency is lower than the IPM, even though in the high-speed operation SRM designs have much better performance. Maximum efficiency of 78.26%, at nominal speed and current, is achieved by 6/10 SRM design. At high current and nominal speed efficiency drops to 71.75%.

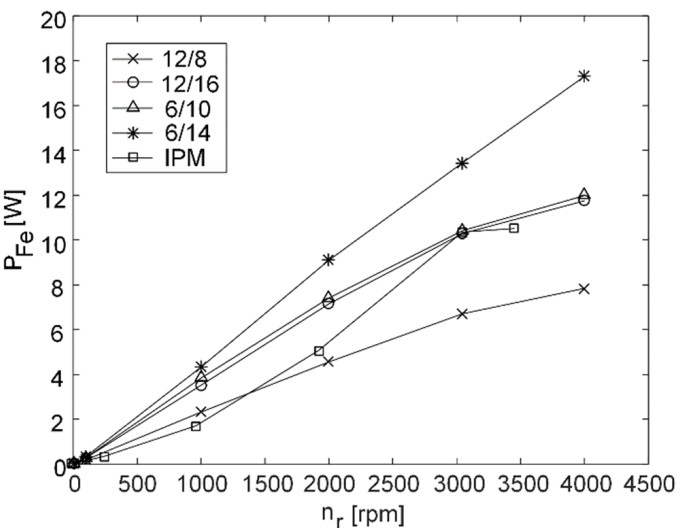

**Figure 19.** Core losses at nominal current operation.

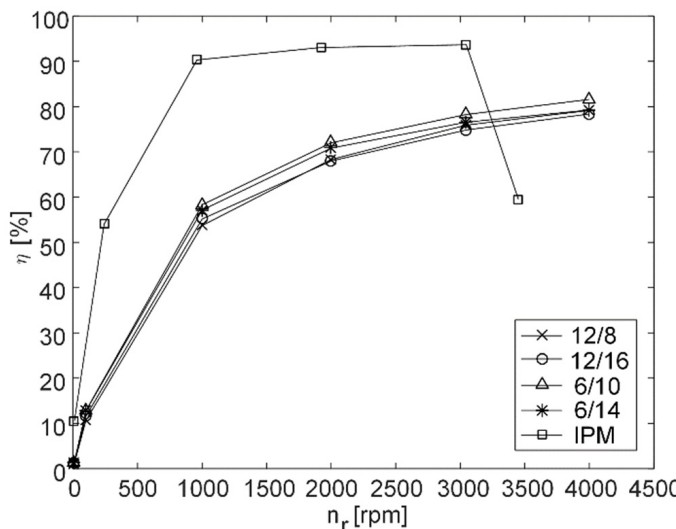

**Figure 20.** Efficiency at different speeds and at nominal current.

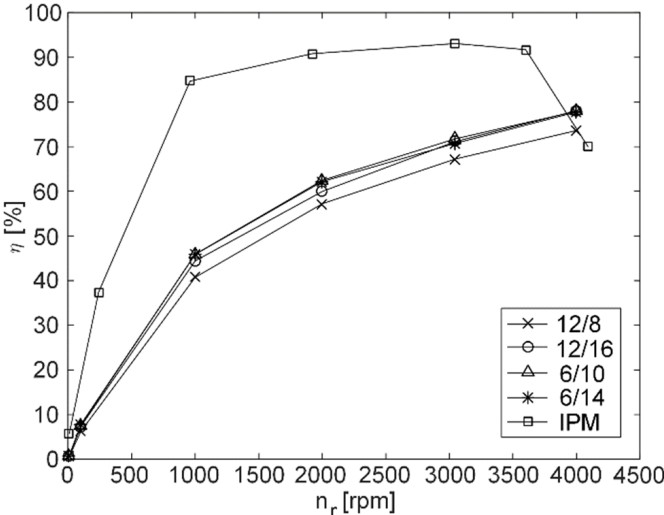

**Figure 21.** Efficiency at different speeds and at maximum current.

Results of thermal analysis are summarized in Figures 22 and 23 for nominal and maximum current, respectively. For any of the designed configurations, maximum permissible temperature is not reached in both the nominal and maximum current operation. The most critical operation points are at low speeds. However, mid drive e-bike motors operate at low-speed region in a very short amount of time, mainly when the pedaling starts [28]. For this reason, even higher currents can be allowed in the SRM winding, which can even further increase the torque density; however, there will be a decrease in the efficiency.

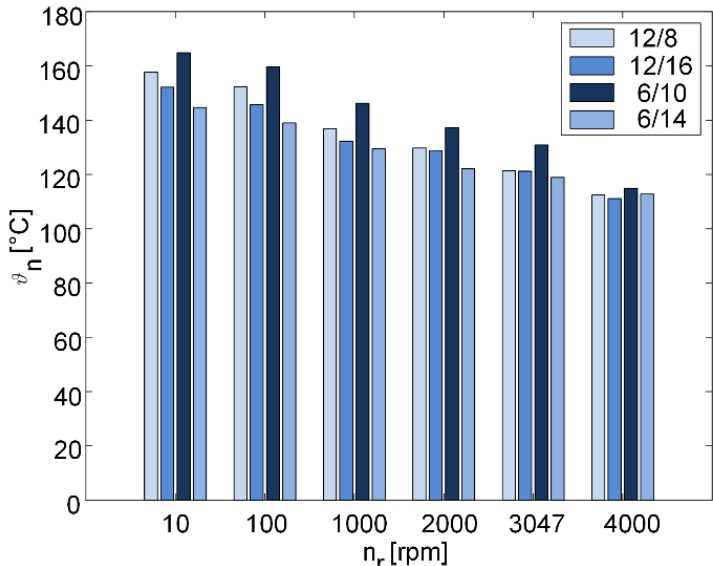

**Figure 22.** Maximum temperature in the motor after 1 h operation at nominal current for different SRM configurations.

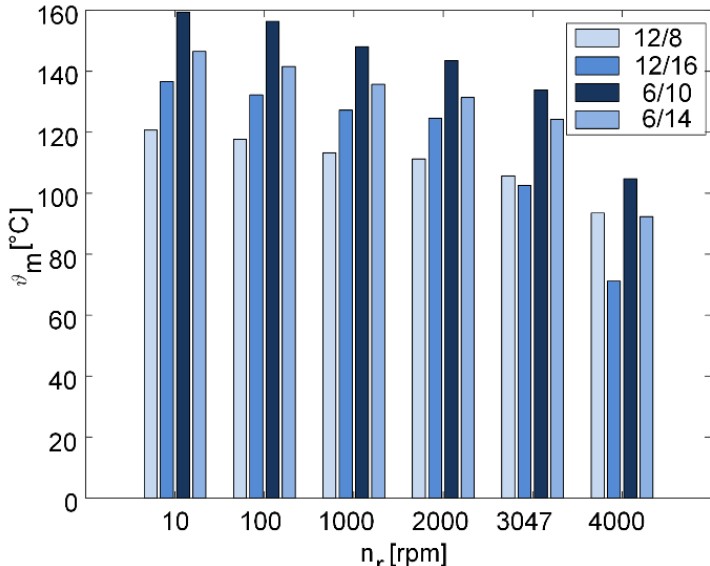

**Figure 23.** Maximum temperature in the motor after 100 s operation at maximum current for different SRM configurations.

### 2.2.4. Selected Design

According to the results presented in the last section, 6/10 SRM is chosen as the most promising design for the given mid-drive application, regarding torque density and efficiency. Among the rest of the configurations, 6/14 is the next one which is preferable,

especially if the torque ripple minimization is the most important design goal. Geometry of optimal 6/10 design is given in Figure 24.

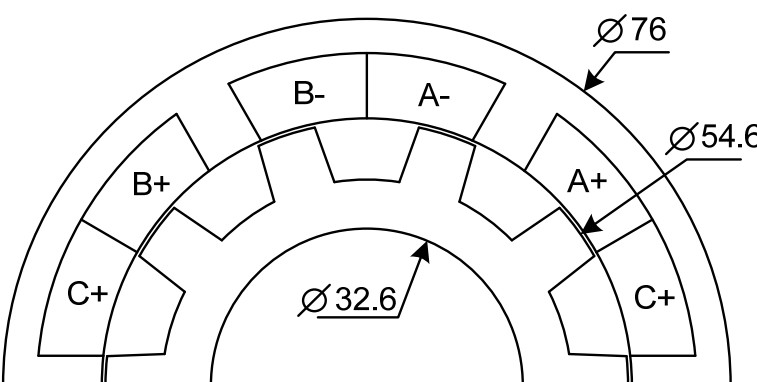

**Figure 24.** Half of optimal 6/10 SRM configuration geometry with main dimensions in millimeters.

## 3. Results of Experimental Verification

This section gives the basic information of the built 6/10 SRM prototype, test bench and experimental results which verify the design procedure and predicted machine performance.

### 3.1. Developed Prototype

Rotor of the final 6/10 SRM design, which is shown in Figure 24, has to be modified for the prototyping to accommodate much smaller shaft which is defined by the maximum torque and mechanical constraints. Modified rotor is shown in the Figure 25. Rotor inner radius is reduced to the size of the shaft and in order to reduce the rotor weight the holes are drilled in the rotor yoke. Electromagnetic simulations are performed as well to see if this modification brings some changes to the torque values. Figure 26 compares static torque curves for the initial and modified rotor designs showing that there is no significant difference between them. It's mainly because the thin 0.7 mm bridges saturate and act like an air in electromagnetic sense. Figure 27 shows the difference between flux linkage curves which is also very small and negligible.

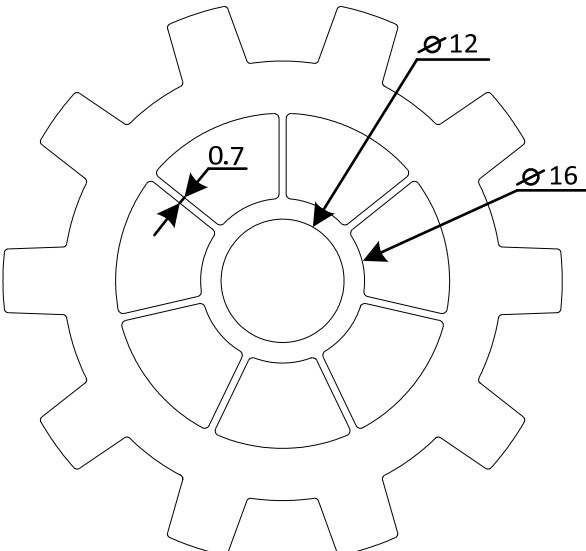

**Figure 25.** Modified rotor geometry for prototyping.

Finally, Figure 28 shows manufactured prototype. Active parts (stator and rotor magnetic circuits and the winding) can be seen on the lefthand side and motor housing on the righthand side.

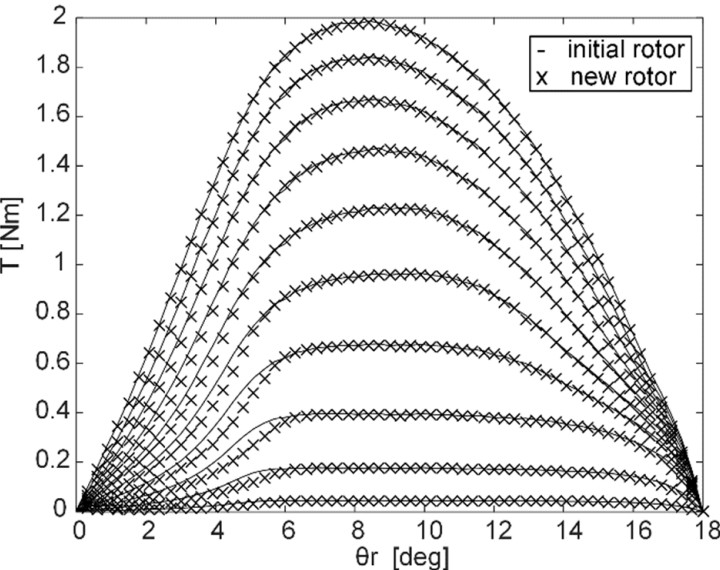

**Figure 26.** Static torque curves for initial and modified rotor designs.

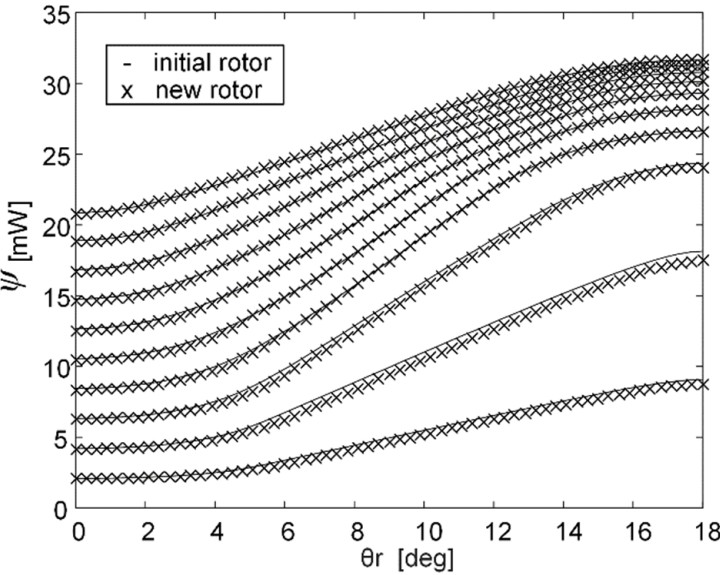

**Figure 27.** Flux linkage curves for initial and modified rotor designs.

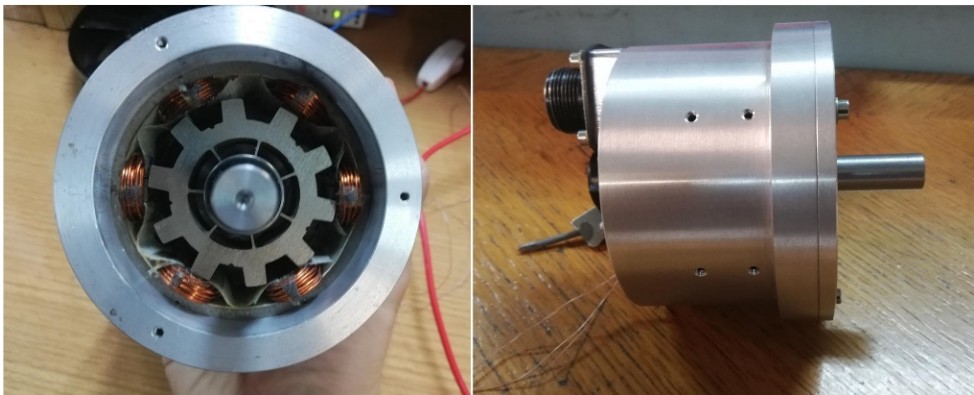

**Figure 28.** Manufactured prototype. Stator with the windings and rotor are shown on the (**left**), and complete motor with the housing on the (**right**) side.

### 3.2. Static Tests

In order to verify electromagnetic design, test bench for determination of static characteristics is built as shown in Figure 29. The test bench contains one perforated disc which is used to position the rotor from aligned (18°) to unaligned (0°) position with the step of 1° mechanical. In each position the rotor is locked, and voltage pulse test is applied on tested phase, while recording the current in the phase. Induced voltage in the successive phase is also recorded for the purpose of determination of mutual flux linkage since it has an impact on dynamic performance of the SRM which is planned to be tested in the future work.

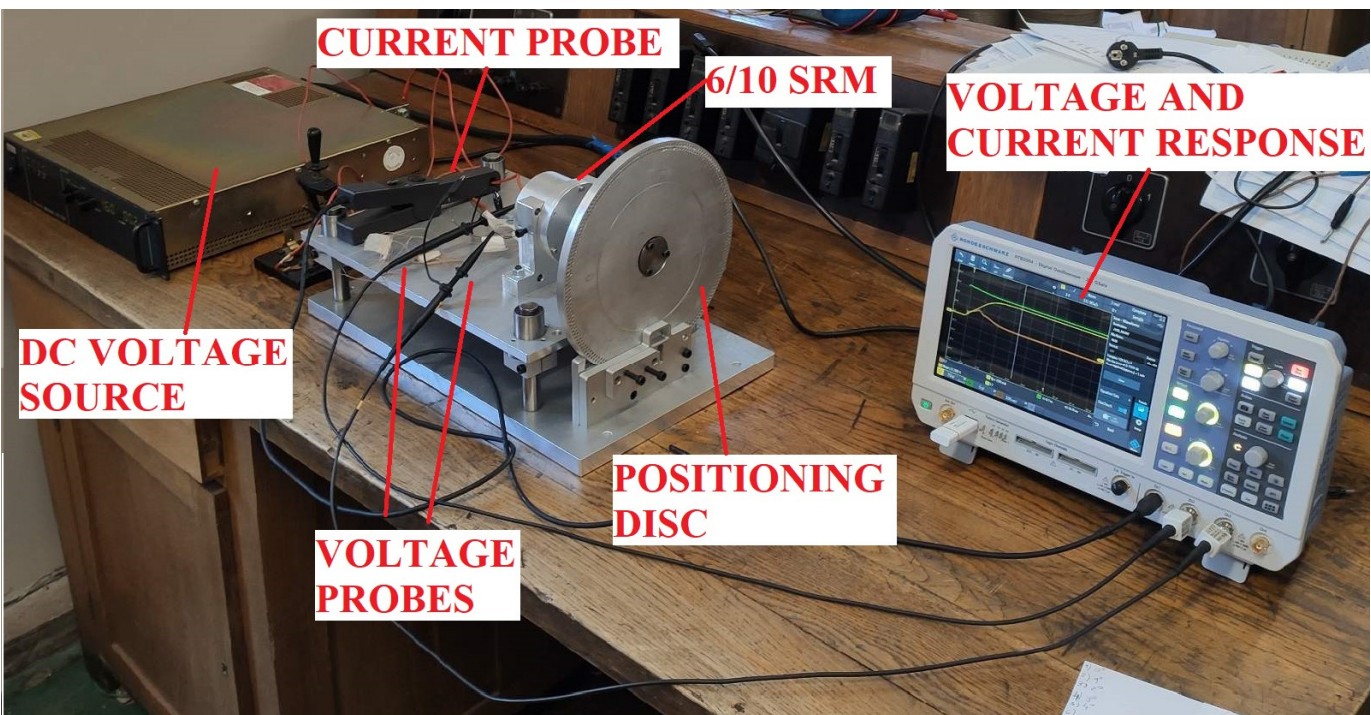

**Figure 29.** Test bench for determination of static electromagnetic characteristics of 6/10 SRM prototype.

Static electromagnetic characteristics, i.e., flux vs. current and position, are determined from current and voltage responses using voltage balance equation for one phase:

$$\psi = \int_{0}^{t_{end}} (u - Ri)dt, \tag{5}$$

where $\psi$ is the flux linkage of tested phase, $u$ is applied voltage pulse, $i$ is current response and $R$ is the winding resistance which is determined from the steady state values of voltage and current:

$$R = \frac{u(t_{end})}{i(t_{end})}, \tag{6}$$

In this way only one measurement was necessary for each position to get flux values for all currents from zero to maximum value of 24 A.

Flux linkage vs. current for different positions is shown in Figure 30, overlapped with the values obtained from electromagnetic calculations. Predicted values match very well with measured values which verifies applied design and optimization procedure.

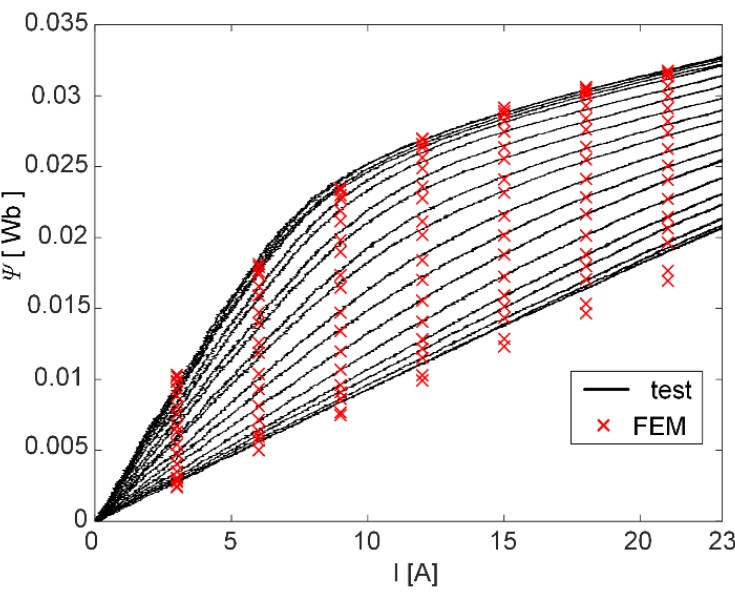

**Figure 30.** Flux linkage curves of 6/10 SRM obtained from static tests and FEM simulations.

### 3.3. Dynamic Tests

Dynamic performance is checked by supplying the prototype from generic lab converter which in this case performs as asymmetrical half bridge converter (AHBC) supplying each phase of the SRM separately. Converter has its own DSP TMS320F28379D control card where closed loop speed control is implemented. Speed is measured via compact incremental encoder which is integrated on the machine housing. Laboratory setup is shown in Figure 31.

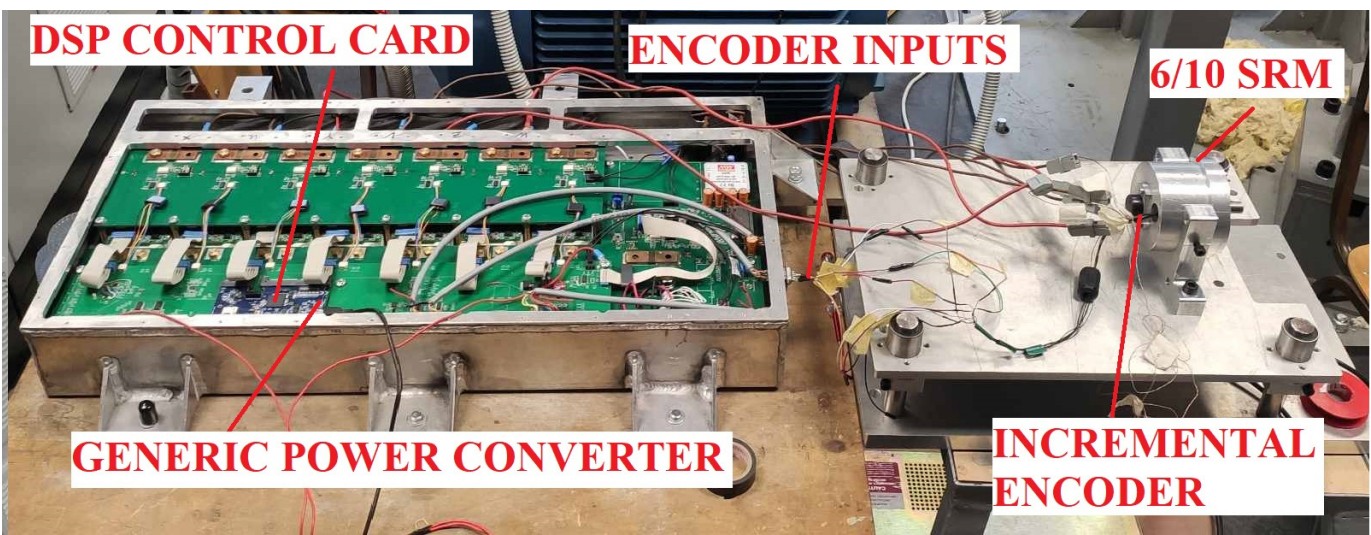

**Figure 31.** Test bench for dynamic no-load tests of 6/10 SRM prototype.

Structure of the SRM control scheme is presented in Figure 32. In order to achieve required SRM performance it is necessary to apply phase currents in accordance with predefined control parameters. This is achieved by means of hysteresis current controller block which takes as inputs chopping current limit ($I_{ref}$), which is the output of the PI speed controller, and optimal turn-ON and turn-OFF angles, which are obtained from the look-up tables (LUTs) [35]. LUTs were calculated using optimization algorithm which goal is to achieve combination of turn-ON and turn-OFF angles which results in maximum torque per ampere for each speed and current in the operation range [36]. According to the measured position, speed is calculated and fed into the PI controller and LUTs block.

Taking $I_{ref}$ and turn-ON and turn-OFF angles as inputs hysteresis current controller gives driving signals for the AHBC converter.

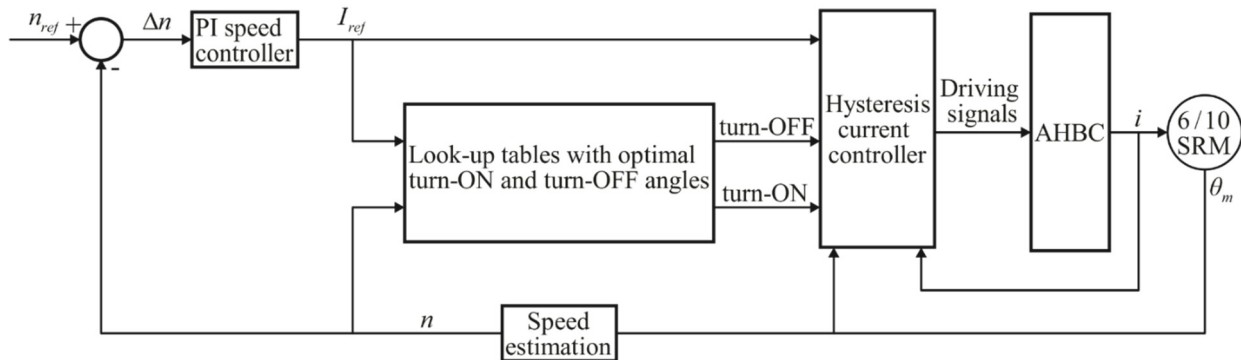

**Figure 32.** SRM control scheme.

The motor is run in no-load condition up to rated speed. DC link voltage is set to 10 V. Currents of all three phases are shown for the speed of 1000 rpm and 3000 rpm in the Figures 33 and 34, respectively. As can be seen motor is in the copping regime at both speeds since the load is very small and corresponds only to the small amount of friction and windage losses. RMS values of phase currents are 1.196 A and 1.343 A for speed of 1000 rpm and 3000 rpm, respectively. Since DC link voltage is set to 10 V, at rated speed motor is close to single pulse regime. Performed test validates electromagnetic and mechanical design and functionality of the motor prototype. Prototype in operation can be seen on video which is included in Supplementary Materials of the manuscript.

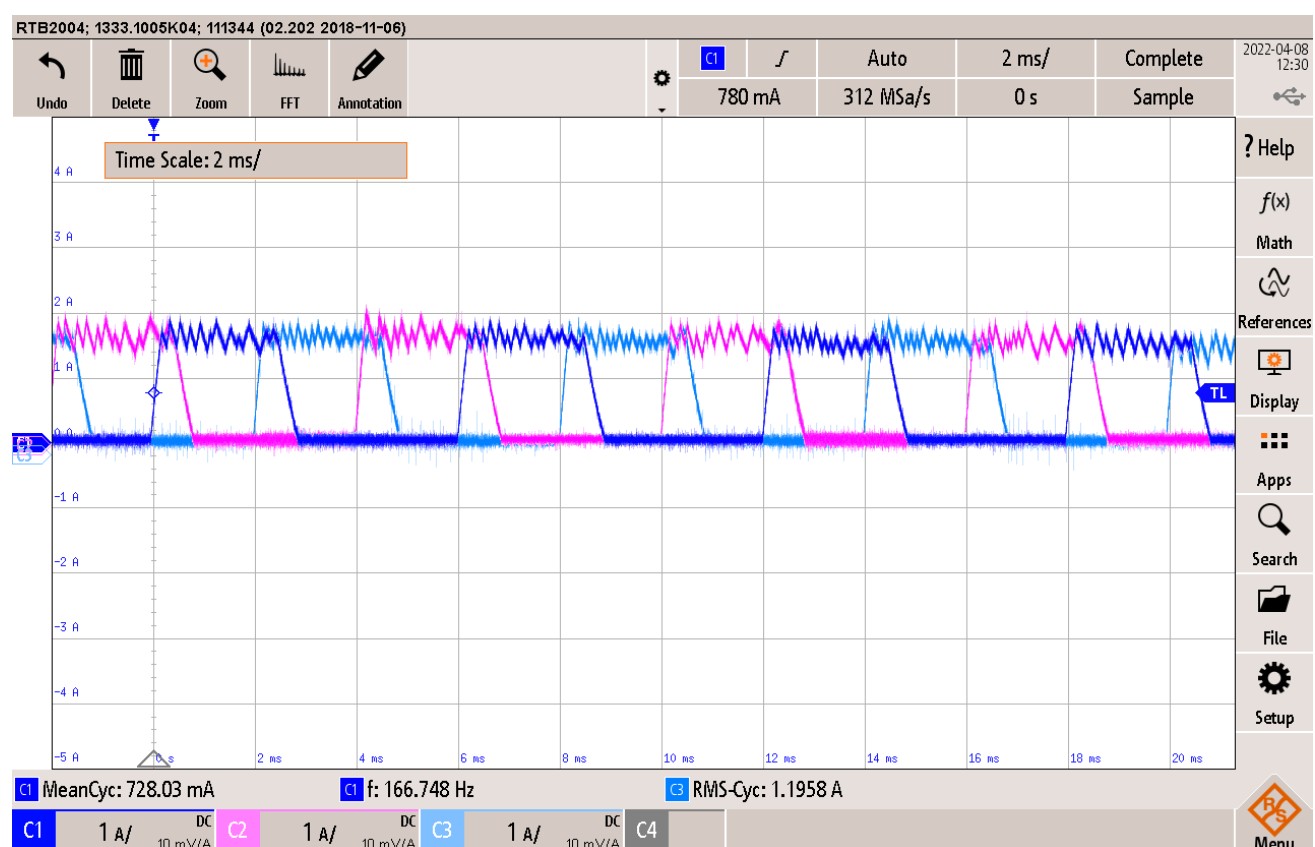

**Figure 33.** Phase currents of 6/10 SRM prototype in no-load operation at the speed of 1000 rpm.

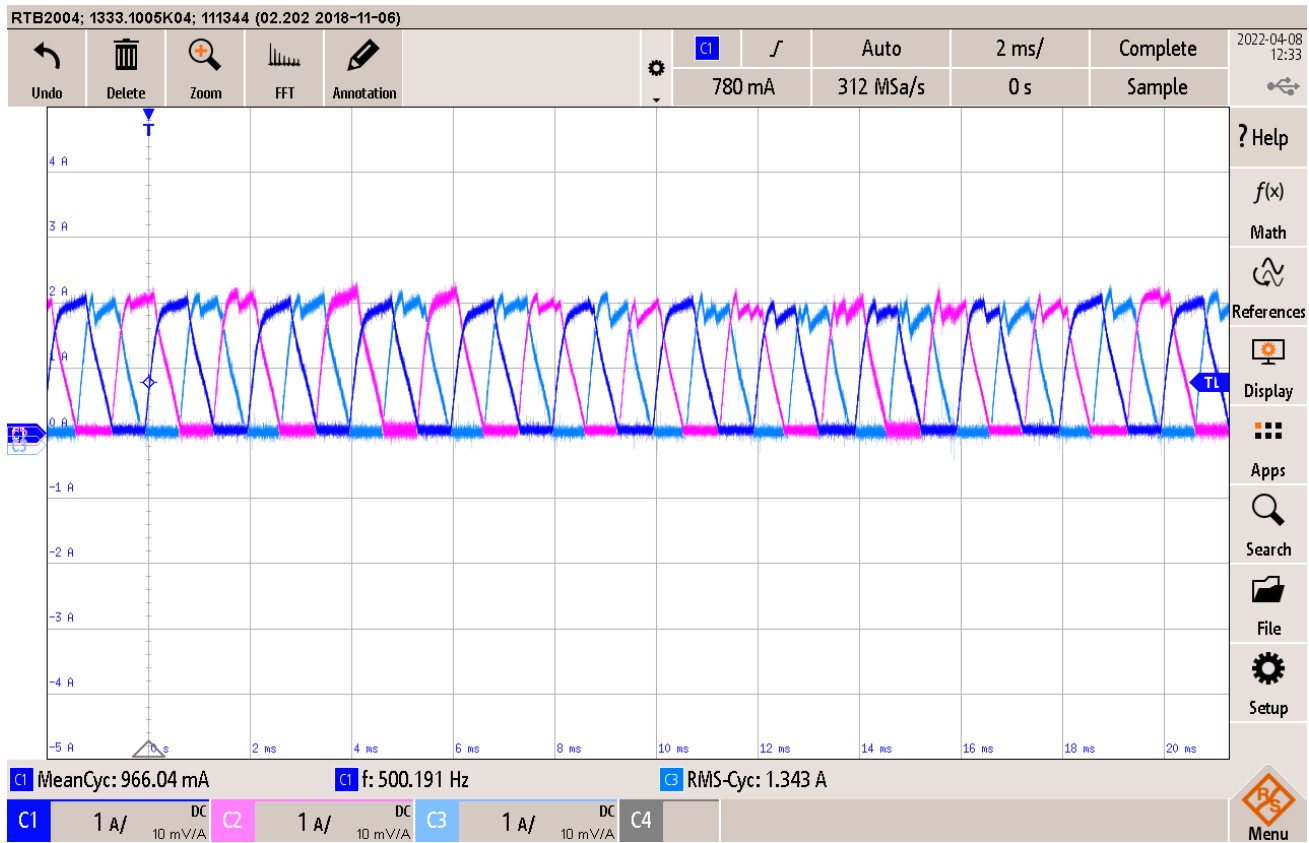

**Figure 34.** Phase currents of 6/10 SRM prototype in no-load operation at rated speed of 3000 rpm.

Additional dynamic performance tests will include measurements of torque-speed curves and efficiency maps, as well as precise rating of the motor by measuring winding temperatures under different loads with built-in thermocouple sensors. It is planned also to use dedicated SRM converter which is currently being developed and optimized for the mid-drive e-bike application.

## 4. Discussion and Conclusions

Through the overview of the existing e-bike mid-drives, presented at the beginning of this paper, their main characteristics are shown and compared. According to this overview, no SRM has been designed and used so far for this application. Thus, the main aim of this paper is to show the SRM design procedure through the development of several SRM configurations: 12/8, 12/16, 6/10, and 6/14 that might be suitable for e-bike mid-drive application.

As a benchmark machine, spoke type IPM from Shimano Steps mid-drive was taken and analyzed in detail. Although this kind of machine has very high/torque density with very high efficiency at the same time, it also has very narrow flux-weakening region which reduces its capabilities to run at very high speeds, as compared to SRM.

By inspecting the obtained results for the designed SRM configurations, we conclude that SRM can be used for this application. Although it has lower torque for the same inverter supply current, there is a possibility to increase inverter rating and SRM phase current, since this kind of machine is very thermally robust. This was also proved through the thermal results, which showed that for the selected H type insulation class there is still large temperature margin for the increase of phase current. In this way, SRM will completely match torque demands which are set by IPM benchmark.

On the other hand, replacing of the IPM with SRM will certainly reduce the cost and weight of e-bikes, which was actually one of the main goals when designing SRM for this application. Also, usage of SRM will increase robustness of the e-bike in harsh environments.

Comparing the different configurations showed that 6/10 has the highest torque density, mostly because of the highest aligned to unaligned inductance ratio. This design result is probably affected by very rigorous space requirements, which have not allowed the change of main motor dimensions. Apart from this, 6/14 SRM also showed very good performance in terms of very low torque ripple. Furthermore, 6/14 has the lowest stator pole/yoke ratio which makes it probably the best solution regarding NV characteristics. Finally, by building and testing the prototype of 6/10, we verified the SRM design procedure and derived conclusions.

**Supplementary Materials:** Included video is available online at https://www.mdpi.com/article/10.3390/machines10080642/s1, which shows prototyped SRM in operation.

**Author Contributions:** Conceptualization and methodology, M.V.T. and D.S.M.; validation, M.V.T. and D.S.M.; writing—original draft preparation M.V.T.; writing—review and editing, D.S.M.; visualization, D.S.M.; project administration, M.V.T. All authors have read and agreed to the published version of the manuscript.

**Funding:** Publishing fee is funded from project at University of Belgrade, School of Electrical Engineering, with internal number 71886 and project leader Mladen Terzić.

**Institutional Review Board Statement:** Not applicable.

**Informed Consent Statement:** Not applicable.

**Data Availability Statement:** Not applicable.

**Acknowledgments:** The authors gratefully acknowledge Kinestas Machines d.o.o for testing equipment used in experimental validation of presented work.

**Conflicts of Interest:** The authors declare no conflict of interest. The funders had no role in the design of the study; in the collection, analyses, or interpretation of data; in the writing of the manuscript; or in the decision to publish the results.

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
