# Peer review of "Switched Reluctance Motor Design for a Mid-Drive E-Bike Application"

_machines, doi:10.3390/machines10080642_

Round 1

Reviewer 1 Report

This paper discusses various SRM design for an e-bike mid drive system. The paper is well written. However, the paper needs improvement.

The author should summarize section 2.1. No need to explain IPM from Shimano Steps mid-drive bike system in detail. The authors should focus on their work. The author should include their optimization procedure.

The authors should also compare the design with BLDC motor, frequently installed in e-bike system.

In figure 12, why the value of torque suddenly fall to low value at 3500 rpm. 

The efficiencies of proposed SRM are less than IPM. How to solve this issue?

What about the noise problem in SRM?

The authors should also include the torque density of proposed SRM design and compared with IPM.

It will be better to installed proposed SRM practically in e-bike and include real data.  

Author Response

This paper discusses various SRM design for an e-bike mid drive system. The paper is well written. However, the paper needs improvement.

Dear reviewer, we appreciate very much that you recognized our efforts, and we thank you for your valuable comments. We did our best to answer all your questions and to improve our paper. Our answers can be found bellow your comments in red and changes in the manuscript are labeled by yellow color.

The author should summarize section 2.1. No need to explain IPM from Shimano Steps mid-drive bike system in detail. The authors should focus on their work. The author should include their optimization procedure.

Thank you for your useful suggestion but we thought that it will be of value to readers to see detail properties of one representative mid-drive motor, which authors couldn’t find anywhere else in the available literature. With all respect to your opinion, we would like to keep all information about detailed IPM Shimano Steps motor analysis for the mentioned reasons.

Moreover, we explained our design and optimization procedure in more detail in the section 2.2.2. The SRM design was carried out using a design-space exploration method, where each geometry parameter combination was swept. Optimization of control parameters (turn-on and turn-off angles and chopping current) was performed using genetic algorithm optimization

The authors should also compare the design with BLDC motor, frequently installed in e-bike system.

Different mid-drive e-bike propulsion systems are compared in Table 1. However, since we only have Shimano Steps mid-drive motor we could only get its full specification and compare it to our design. We take Shimano Steps motor as a fair representative of all mid-drive e-bike solutions and took it as a benchmark for the design of SRM motor variants.

In figure 12, why the value of torque suddenly fall to low value at 3500 rpm. 

The value of the torque falls suddenly because the motor entered flux weakening region which is for the spoke type motor very short, due to the fact that the difference between Ld and Lq inductances is negligible, as can be seen from Table 3 (the last row). The difference of Ld and Lq defines the capability of the motor to produce reluctance torque and to extend the operation beyond the base speed at which the maximum voltage available has been reached.

The efficiencies of proposed SRM are less than IPM. How to solve this issue?

This can be solved only by increasing the dimension of the copper conductors and thus the overall size of the SRM. But since the main goal of this work was to achieve the best possible performance from the same dimensions as IPM the efficiency could not be additionally increased.

However, even with the bigger cross section of the copper wires the efficiency of the SRM drive will still remain below the efficiency of the IPM due to the fact that SRM needs to compensate the lack of permanent magnet’s magnetomotive force (MMF) with increased MMF from the stator winding, which means more copper and/or higher current which leads to higher stator copper losses and thus lower efficiency.

What about the noise problem in SRM?

We didn’t tackle this broad topic in this work since our main goal was to achieve the best performance (torque per amp with minimum torque ripple) out of the given space. This would require more extensive research which will be covered in our work in the future.

Generally, SRM has higher noise level than IPM because that is inherent property of the SRM operation and very rich spectrum of radial forces. Nevertheless, we were positively surprised by acoustic performance of our prototype, as you can see in included video.

The authors should also include the torque density of proposed SRM design and compared with IPM.

Thank you for this comment. This is certainly valuable information to show in the paper for the comparison of different variants. We included this information in Table 3 and Table 6 in our revised manuscript.

It will be better to installed proposed SRM practically in e-bike and include real data.  

Yes, it would be nice to see that but currently we have no possibilities to do this since some mechanical parts need to be remanufactured and also dedicated power electronics board to drive the SRM, at which we are currently working on. The final goal is to install the complete drive and to test it in a real environment.

Reviewer 2 Report

The paper discusses the design of a switched reluctance motor (SRM) for an e-bike application. A comparison is made with a commercially available interior permanent magnet motor (IPM). An experimental verification of the SRM model is provided. The reviewer appreciates that the authors have done a great deal of practical work. At the same time, many claims of the authors look questionable:

1) Please add in the introduction the required torque-speed curve for the e-bike motor application in question. What constant power speed range (CPSR) is required?

2) The reviewer believes that those shown in fig. 12-14 comparative characteristics are not correct. It is well known that when applied with a field weakening strategy and properly designed, the IPM provides a CPSR of at least 1:5 [doi.org/10.1109/ACCESS.2019.2950773]. It is not clear why, in the results obtained by the authors, the IPM power decreases so rapidly as the speed increases. What field weakening strategy was adopted in the IPM simulation?

3) The authors write: “Furthermore, its simple and robust structure, as well as low production cost gives SRM an advantage over permanent magnet motors”.

The reviewer agrees that the SRM magnetic core has a lower cost than permanent magnet motors. However, the SRM requires an inverter with a significantly higher maximum current than a comparable permanent magnet motor. Therefore, it is necessary to compare in detail not only the cost of motors, but the maximum currents and the cost of inverters for the SRM and for IPM.

4) The authors write: “Moreover, utilization of SRM configuration with higher number of rotor poles than the stator poles offers higher power density and lower torque ripples”.

It seems that the authors here claim that the SRM has less torque ripple and better specific characteristics than a synchronous permanent magnet motor. However, this contradicts the point of view accepted in the literature. The point of view of the authors requires serious justification. It is well known that motors without magnets have lower specific torque than motors with high coercivity permanent magnets [doi.org/10.3390/wevj13040057].

5) What is the acoustic noise of the proposed SRM? Acoustic noise is known to be a big problem for SRMs. It is necessary to compare the acoustic noise of the SRM and IPM.

Author Response

The paper discusses the design of a switched reluctance motor (SRM) for an e-bike application. A comparison is made with a commercially available interior permanent magnet motor (IPM). An experimental verification of the SRM model is provided. The reviewer appreciates that the authors have done a great deal of practical work. At the same time, many claims of the authors look questionable:

Dear reviewer, thank you very much for your valuable inputs and efforts you put to review our work presented in the manuscript. We also appreciate that you recognized great deal of work we put into this project. We did our best to answer all your questions and to improve the quality of our manuscript.

1) Please add in the introduction the required torque-speed curve for the e-bike motor application in question. What constant power speed range (CPSR) is required?

This information is added into section 2.2.1 which shows other design constraints. However, this curve is also included in the Figure 12 for the nominal and maximum operation requirements. There is no special requirement for the SPSR since the benchmark IPM has very short flux weakening region, so the main goal was to maintain the same power at least up to 4000 rpm. This property is much better in the SRM than in benchmark IPM because it is a spoke type which has negligible difference between Ld and Lq inductances (see Table 3, the last row) and thus very short flux weakening region.

2) The reviewer believes that those shown in fig. 12-14 comparative characteristics are not correct. It is well known that when applied with a field weakening strategy and properly designed, the IPM provides a CPSR of at least 1:5 [doi.org/10.1109/ACCESS.2019.2950773]. It is not clear why, in the results obtained by the authors, the IPM power decreases so rapidly as the speed increases. What field weakening strategy was adopted in the IPM simulation?

We agree with your remark that IPM generally have much longer SPRS but Shimano Steps IPM is a spoke type which has much smaller difference between Ld and Lq (can be seen in Table 3, last row)than the IPM which is given in doi.org/10.1109/ACCESS.2019.2950773. This difference determines the capability of the motor to work in the flux weakening region and under CPSR. Since, the motor from the given reference has a V-shaped geometry of the magnets it has much higher Lq than Ld which gives the motor additional reluctant torque component.

3) The authors write: “Furthermore, its simple and robust structure, as well as low production cost gives SRM an advantage over permanent magnet motors”.

The reviewer agrees that the SRM magnetic core has a lower cost than permanent magnet motors. However, the SRM requires an inverter with a significantly higher maximum current than a comparable permanent magnet motor. Therefore, it is necessary to compare in detail not only the cost of motors, but the maximum currents and the cost of inverters for the SRM and for IPM.

Our sentence was not completely clear and we changed it in revised version of the manuscript. If we observe just the motors (SRM and IPM) in the complete drive than the price of the materials and manufacturing is lower for the SRM because it doesn’t have magnets. On the other side, the SRM converter is usually more expensive than the IPM inverter due to higher number of semiconductor devices and larger DC link capacitance.

However, it is hard to give a general comparison because it highly depends on the application for which the drive is designed for. Furthermore, the converter and the SRM must be designed together in order to achieve optimal drive for the set of design goals and restrictions.

4) The authors write: “Moreover, utilization of SRM configuration with higher number of rotor poles than the stator poles offers higher power density and lower torque ripples”.

It seems that the authors here claim that the SRM has less torque ripple and better specific characteristics than a synchronous permanent magnet motor. However, this contradicts the point of view accepted in the literature. The point of view of the authors requires serious justification. It is well known that motors without magnets have lower specific torque than motors with high coercivity permanent magnets [doi.org/10.3390/wevj13040057].

This has been a misunderstanding due to badly written sentence. The intention was to say that SRM configurations with higher number of rotor than stator poles (for example 6/10, 12/16 etc.) have higher torque density and lower torque ripple compared to SRM configurations with lower number of rotor than stator poles (for example 6/4, 12/8 etc.) and not compared to IPM.

We corrected this sentence in the revised manuscript.

5) What is the acoustic noise of the proposed SRM? Acoustic noise is known to be a big problem for SRMs. It is necessary to compare the acoustic noise of the SRM and IPM.

We agree with reviewer comment but since the main goal of our work was to design SRM for maximum torque per ampere with very strict space requirements (the same volume as IPM) we couldn’t include NVH into optimization process because it would definitely require higher stator yoke and overall volume. However, we took care to minimize torque ripple, as one of the sources (not the main) of noise. The topic of designing an NVH compatible SRM is totally in contrary to our design task we had here, because NVH respectful SRM design leads eventually to the design with much smaller torque density as can be concluded from [1]. On the other side, there are some current shaping techniques for suppressing some components in radial force spectrum in order to alleviate some resonances and we plan to work on these techniques if we find it necessary. However, during the testing so far, we haven’t experienced any noise issue (we haven’t measure it though because we didn’t have equipment), and attached video can witness that.

[1] T.J.E.Miller, Electronic Control of Switched Reluctance Machines (Chapter 4-Design for low noise), Newnes; 1st edition (July 9, 2001), ISBN-10:0750650737, ISBN-13:978-0750650731

Round 2

Reviewer 1 Report

Thanks for revising the paper.  No further comments.

Reviewer 2 Report

Thanks to the authors for the detailed answers. The paper may be published.